# Can Data Videos Influence Viewers' Willingness To Reconsider Their Health-Related Behavior? An Exploration With Existing Data Videos

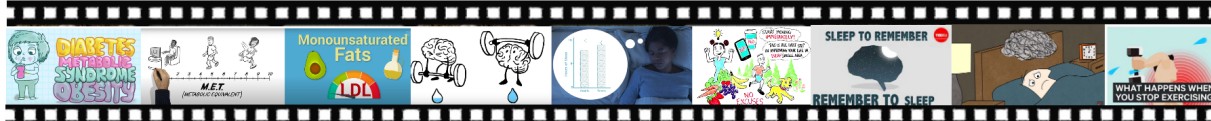

Figure 1: Data Videos or animated infographics are short videos that present large amounts of data in an engaging narrative format. We selected nine Data Videos focusing on three health-related topics; physical activity, sleep, and diet. This figure shows an epitome of the videos used in the study. For the full list of videos, their links, and more detailed description, refer to Appendix A, B.

## ABSTRACT

Data Videos have the potential to promote healthy behaviors [21]. Using publicly available data videos addressing physical activity, sleep, and diet, we explored the persuasive capability of Data Videos through their narrative format and the affective connection they arouse in their viewers' minds. We asked four central questions; (1) do Data Videos increase negative affects in their viewers?; (2) are negative affective responses linked with individuals' personality traits?; (3) can negative affects predict any change in viewers' willingness to improve their health-related behavior?; and finally (4) can personality traits and/or video attributes predict viewers' such willingness? An M-Turk study was conducted, whereby participants ($N = 102$) watched Data Videos, answered questions about their perceptions, and completed a personality trait questionnaire. Overall, influencing participants' willingness to reconsider their health-related behaviors was more difficult (i.e., harder to persuade) when they scored higher on neuroticism. This was because these individuals liked the provided Data Videos less, compared to those who scored low in neuroticism, at least partially. Perceived usefulness of the information along with neuroticism predicted our participants' willingness to reevaluate their health-related behaviors. Together, these findings show the importance of both using personality traits (i.e., personalization) and working on the general contents of Data Videos without considering personalization.

**Index Terms:** Human-centered computing—Human computer interaction (HCI)—Empirical studies in HCI

## 1 INTRODUCTION

Behavioral health refers to how individuals' well-being and health are influenced by their own behaviors [2]. Simple and preventable as they sound, behavioral health issues like improper diet or physical inactivity could be linked to many serious consequences such as cardiovascular diseases, obesity, high blood pressure, and even some types of cancer [1, 61]. In fact, not engaging in a sufficient amount of physical activity is a key risk factor for death worldwide [1]. In order to find means to effectively address our behavior-related health challenges in a timely manner, research involving technology will be key.

As behaviors are normally adjustable, people suffering from behavior-induced health issues would benefit if they understood how their seemingly minor behaviors are tied to their serious medical issues. Further, they could be motivated to reevaluate their behaviors if they were guided with concrete behavioral suggestions on how

to improve their health. The availability of modern technologies in the form of wearable devices and mobile health apps (mHealth) allows us to collect diverse personal health-related data (e.g., steps taken, calories burnt, and sleep hours) [51]. However, users of these technologies often do not benefit fully from the data they obtain [24]. This is because current data representation approaches generally lack the ability to "convey" the insights to users so they can take action. While current mHeath technologies typically provide statistics in the form of charts and graphs [39, 57], research shows that such methods are less effective in triggering behavior change [46], which is the ultimate goal of such technologies. Statistical data representation of personal health data is often passive, difficult to interpret, and not insightful. Thus, users often do not explore these representations fully. Some even claim the way such personal data is presented could be too frustrating or overwhelming for the users to understand [57]. This frustration could then lower the users' motivation to reexamine their health-related behaviors, and even their motivation to continuously use the technology [57]. We believe that the role of modern health tracking technology could move beyond data collection and presentation, towards presenting data in an insightful way to provide "*actionable intelligence*" to effectively guide users.

For the effective delivery of health information, we focus on Data Videos in this study. Data Videos or animated infographics are short in length, typically shorter than six minutes [36], and provide factual, data-driven information for the users in an engaging *narrative format* [6, 7, 47]. The narrative nature of Data Videos can make complex data easier to digest and act upon because narratives are a natural way for people to communicate and gain knowledge, [7, 21]. As Data Videos take a storytelling format, they arouse emotional connections in their viewers' minds while they engage with the story. Research in advertisement and marketing demonstrates that *affect* plays an important role in motivating and convincing viewers and consumers [8, 50, 52]. Based on these findings, we hypothesize that a potential strength of Data Videos could be attributed to their capability to rouse the viewers' affect (e.g., people realize how much sugar they consume everyday, and they become motivated to reduce their daily sugar intake because they *fear* negative consequences). Indeed, to alleviate negative affects, changing one's behavior is logical (e.g., I don't want to get sick so I will start exercising). In this way, our affects could be indirectly driving our behaviors. Further, personality differences could play an important role in individuals' emotional responses [13, 35, 42] as well as their potential behavior modification in response to a persuasive system [9, 37, 40, 44, 63, 77]; individuals' responses to an emotion provoking persuasive message vary from person to person. Accordingly, we plan to consider personality differences as a factor in our study.

Another aspect that can play an important role in the persuasive capability of health Data Videos is how the viewers perceive the

content of the video (i.e., their content value appraisal). Could they follow the content of the video easily and clearly? Did the video provide them with new and/or useful information?

The problem we are addressing in this paper is how to improve the persuasive potential of health-related Data Videos, to effectively influence viewers' willingness to alter their health-related behavior. Our focus is in on two dimensions: 1) personalizing Data Videos to viewers' unique personalities, and 2) improving the overall quality of Data Videos in general.

## 2 RELATED WORK

We discuss previous work investigating the effectiveness of Data Videos as a narrative to communicate data. We highlight some behavioral theories as well as studies in HCI/other related fields, then turn to affects associated with narratives. Finally, we discuss how affects play a role in forming people's attitudes. As a factor influencing affects, we review studies focusing on the importance of the role personality traits play in reaction to a persuasive message, by describing studies and strategies used in the persuasive technologies literature.

### 2.1 Data Videos as a Narrative

Data videos are motion graphics that incorporate factual, data-driven information to tell informative and engaging stories with data [5, 6]. Data videos are gaining popularity [6] in various fields such as journalism, education, advertisements, mass communication, as well as in political campaigns [29, 38, 47, 70–72]. Due to their narrative nature, Data Videos are recognized as one of the seven forms of narrative visualization [19, 71]. Baber et al. [10] define narrative as a formal structure that constitutes a "sharable" story as opposed to the informal stories which could be "unstructured" and "ambiguous". A narrative is a series of connected events that constitute a story [71]. The order in which these events is presented in a medium constitutes its narrative structure [5]. Amini et al. [6] examined 50 professionally created Data Videos to learn about their narrative structure. In their study, they divided the videos into temporal sections and coded them based on Cohn's [23] theory of visual narrative structure that categorized the narrative into four stages: Establisher (E), Initial (I), Peak (P) and Release (R). Amini et al. [6] provided insights regarding the average duration (in percentage) each narrative stage consumes from the total video length as well as the percentage of time spent on attention cues and data visualizations within each stage. They also pinpointed some narrative structure patterns that are commonly used in Data Videos.

The power of Data Videos comes mainly from this narrative format. Stories can convey information in an engaging way that is more natural, seamless, and effective than text or even pictures [31, 39]. A well told story can convey a large amount of information in a way that the viewers find interesting, easy to understand, trust, recall readily, and make sense of [17, 31, 57]. The advantage of visual narrative is its ability to present plenty of information in a compact form, compared to text or pictures alone [31]. According to Narrative Transportation Theory, videos can transform and immerse the viewer in a totally different world with their locale, characters, situations, and emotions which could reflect on the users' own beliefs, emotions, and intentions [34, 57, 58, 76]. Furthermore, a plethora of psychological theories support the persuasive power of narrative. The Extended Elaboration Likelihood Model (E-ELM) argues that as people indulge in a narrative, with all its cues and stimuli, their cognitive processing of the narrative obstruct any counterarguments of the presented message [18, 74], making the message more persuasive even for those who are difficult to persuade otherwise [73]. Furthermore, as per the Entertainment Overcoming Resistance Model (EORM), the entertaining aspect of a narrative also plays a role in reducing the cognitive resistance to the message presented, and hence facilitates persuasion [22, 56, 57].

Despite the great potential of, and the increasing demand for, Data Videos for information communication, it was not until recently that researchers turned an eye to empirically investigate them in terms of their building blocks, components, and narrative characteristics [6]. In a recent study that aimed at exploring the persuasive power of Data Videos, Choe et al. [21] introduced a new class of Data Videos called Persuasive Data Videos or PDVs [21]. This genre of Data Videos incorporates some persuasive elements inspired by and drawn from the Persuasive System Design Model [62]. In their research, the authors studied how incorporating some persuasive elements in a Data Video could improve the potential persuasion level of the video [21]. Their study revealed that their PDVs had higher persuasive potential than regular Data Videos.

Amini et al. [7] examined the effect of using pictographs and animation, two commonly used techniques in data videos [7]. They found that the use of such techniques enhanced the viewers' understanding of data insights while boosting their engagement. They concluded that the strength of pictographs can be attributed to their ability to trigger more emotions in the viewers, while the animation strengthens the intensity of such emotions.

### 2.2 Affects and Data Videos

This leads us to an important aspect of Data Videos: affects. Research shows that viewers' preference for multimedia; be it a performingart, internet video, or even music videos, is highly dependent on their arousal level and the intensity of their affects towards the viewed media [11, 75]. Past studies assumed that TV viewers liked to watch shows that elicit positive emotions as opposed to negative emotions [32]. However, later research showed this may be true for real life events, but people enjoy watching TV shows that evoke fear, anger, or sad emotions [59]. Bardzel et al. [11] examined the intensity and valence of viewers' affects, as well as their ratings of internet videos. The results showed correlation between the affects' intensity and the liking of the video [11]. As for the valence (i.e., positive or negative), the study showed that it is not the presence or absence of certain affects, be it negative or positive, that influenced the rating of the video. It is rather the *emotional arc* that leaves the viewer emotionally resolved and hence liking the video, even if it started with negative emotions [11]. In fact, health-related videos and campaigns are often designed in a way that elicits negative affects such as fear, worry, anxiety, etc. This is because health promoting messages normally present the negative consequences of not following a healthy behavior or of engaging in an unhealthy behavior (e.g., if you do not exercise you will become obese, look older, be at risk of diabetes, high blood pressure and cancer, or if you smoke, you will look like the picture on the tobacco box), as well as alarming statistics on how many people are suffering from those consequences. Such strategies have been studied, approved, and even recommended for public health campaigns such as anti-smoking campaigns [14, 64]. Theoretical studies in health risk messaging design suggest that a certain level of threat is "required" for the message to be effective, while excessive levels of threat could backfire [55]. This is likely the reason that Health-related Data Videos frequently contain alarming messages.

#### 2.2.1 The role of affects in attitude and behavior change

Affects play an important role in the appeal, as well as the persuasive power, of media [3, 15, 50]. According to behavioral theories in psychology, some of our attitudes have a cognitive basis while others have an affective basis [45, 65]. *Affective attitudes* emerge from our feelings towards certain topics or ideas. Some attitudes are influenced relatively easily through affects or emotions while others through logic and facts [45, 65]. The Dual Process Model suggests two routes to persuasion; central and peripheral. The central route is the cognitive route in which the receiver of a message is willing and able to cognitively process the ideas [18, 65]. In contrast, the

peripheral route processing is triggered when the receiver lacks the motivation or ability to logically process cues in the message, and decides to agree with the message based on its emotional appeal (e.g., emotions triggered by the look or smell, but not by the logic) [18, 50]. For instance, one might purchase a car based on its gas emission, cost, functions, and so on (Central route) or because of the way it looks (Peripheral route). In sum, research indicate both cognition and affect are heavily involved in persuasion.

In the field of marketing, as an area focusing primarily on persuading and guiding the viewers to adopt a certain service or commodity, a wide array of studies focused on the kind of affects evoked by ads [41] and how they affect the viewers' attitudes to improve the persuasive power of ads [16]. Models for behavior change, such as the health belief model [43, 68] support that negative affects (e.g., feeling worried or at risk) are the first step towards behavior modification. That is, people need to recognize they are at risk in order to be willing or motivated to change their behavior. Dunlop et al. [27] examined the responses to health promoting mass-media messages and found that feeling at risk was a significant predictor for participants' intention to attitude change. Here, we should say "Additionally, partly related to this mode, health-related Data Videos almost always include some negative information (e.g., Negative outcomes of lack of sleep).

### 2.2.2 Measuring Affects

There is a wealth of research in diverse fields such as psychology, advertising, political science, and HCI on measurement of viewers' affective responses to videos, advertisements, or computing applications. When it comes to measuring affects, studies normally fall between two approaches or a mix of both. The first approach is the implicit approach to measure affects, which relies on physiological recordings of individuals' biometric responses. The second approach is the explicit self-reporting of the viewers' or users' affects during their exposure to the stimuli. While modern technologies in the form of sensors and specialized devices that can log biometric changes (e.g., heart rate, breath rate, respiration patterns, skin patterns, electroencephalogram (EEG), and galvanic skin responses or GSR) are very promising [49, 75], they are often invasive, expensive, and the meaning of the data recorded remains unclear [49]. As for the explicit measurement methods, indicators such as final applause to a show, post-show surveys, or interviews are most commonly used. While less costly and invasive, the explicit approach could capture somewhat skewed responses as the viewers' responses are normally affected by their peak emotion and the emotions experienced at the end of the show (i.e., the 'peak-end' effect) [25, 48, 49, 67]. More recent research relies on participants' self-recordings of their affects using different forms of sliders. Latulipe et al. [49] developed two self-reporting scales; the Love-Hate scale (LH scale) and the Emotional Reaction scale (ER scale) [49]. The LH scale was implemented on a slider that had the labels 'Love it' and 'Hate it' at the very ends and neutral in the middle. The ER slider, on the other hand, ranged from 'No Emotional Reaction' to 'Strong Emotional Reaction'. Researchers in this study wanted to relate self-reported emotions recorded by participants while watching a video, using one of the LH and ER scales, to biometric data collected using GSR, and this is indeed what they found: they found strong correlation between ER scale and GSR ($r = .43$; $p < .001$). The absolute value of the LH scale was also strongly correlated with GSR data. In this study and similar studies, the researchers used a continuous reporting of emotions where participants rated their emotions all through the video. Another approach can be by chunking the video into meaningful segments and reporting emotions in each segment [49].

### 2.3 Personalization in Persuasive Technology

Recent research indicates that the one-size-fits-all model of persuasive technology is not as effective for persuading users to change attitudes or behavior. Instead, the focus is shifting towards *personalized* persuasive systems which often explore the effect of personalities on persuasion level [9, 20, 26, 40, 44, 45, 77, 79]. The five-factor (or Big Five) model of personality offers five broad personality traits: extraversion, agreeableness, conscientiousness, neuroticism, and openness to experience. This is the most widely used model for personality assessment across diverse disciplines [28, 30, 45, 53, 69, 77], it has been repeatedly validated, and its predictive power has been confirmed [82]. Halko et al. [37] explored the link between personality traits and people's perception regarding persuasive technologies that adapt different persuasive strategies. They categorized their participants based on the Big Five personality traits, and studied the effect of eight persuasive techniques. These eight techniques were grouped into four categories by putting complementary persuasive strategies together. The Instruction style category, for example, included authoritative and non authoritative, Social feedback included cooperative and competitive, Motivation type included extrinsic and intrinsic and the Reinforcement type included negative and positive reinforcement. They found correlations between personality traits and the persuasive strategies. For example, their results showed that people who score high in neuroticism tend to prefer negative reinforcement (i.e., removal of aversive stimuli) in the reinforcement category. As for the social feedback, neurotic people do not prefer cooperating with others to achieve their goals.

## 3 STUDY DESIGN

To reiterate, we explored the influence of existing health-related Data Videos on users' willingness to reconsider their health-related behavior. We focused on three health-related topics (physical activity, sleep, and diet). We examined personality differences as a factor and whether personality contributes to the affective experience of viewers watching Data Videos. We also examined some factors related to participants' appraisal of the videos' content in relation to their potential to change their attitude. For this exploration, we developed an online study which contained questions and the Data Video stimuli.

### 3.1 Study Administration

An online study was created using Qualtrics, and administered through Amazon Mechanical Turk (M-Turk). To ensure data quality, we recruited participants with approval ratings higher than 95% and who had completed a minimum of 1000 tasks prior to our study. All participants received monetary compensation ($2.24 US) in compliance with the study ethics approval and M-Turk payment terms. We restricted participant recruitment to the US and Canada to help ensure a good command of English. The study started with a consent form, and provided participants an overview of the objective of the study, and the study instructions. The study consisted of survey questions before and after the presentation of three Data Videos on health-related topics.

### 3.2 Data Video Selection

We collected Data Videos focusing on three general health-related topics; physical activity, healthy sleep, and healthy diet. Our aim was to collect generally *good* Data Videos as our ultimate goal is to create guidelines to produce Data Videos. There are no criteria for quality of Data Videos yet: empirical research in this area is scarce, so we systematically explored existing Data Videos with guidance from Amini et al.'s [6] study. Our careful video selection process, described below, yielded overall consistency across all the videos used (See Fig. 5).

- First, two researchers collected more than 100 Data Videos using relevant keywords such as 'healthy diet', 'dangers of not having enough sleep', 'importance of exercise', etc.

Table 1: Big Five Inventory 10-items (BFI-10) developed by Rammstedt [66]

| I see myself as someone who ... | |
| --- | --- |
| ... is reserved (R)
... is outgoing, sociable | *Extraversion* |
| ... is generally trusting
... tends to find fault with others (R) | *Agreeableness* |
| ... tends to be lazy (R)
... does a thorough job | *Conscientiousness* |
| ... is relaxed, handles stress well (R)
... gets nervous easily | *Neuroticism* |
| ... has few artistic interests (R)
... has an active imagination | *Openness* |

*(R) = item is reverse-scored.*
A Likert scale (1: Strongly Disagree to 5: Strongly Agree) is used.

- We then removed videos that did not follow the Data Video definition found in [6] or contained erroneous information.

- Remaining videos were coded by two researchers for length, source credibility, information accuracy, etc. To ensure the quality and accuracy of the information provided in the videos, we increased the score of videos produced by professional and reputable health-related organizations, companies, magazines, research centers, and websites (e.g., WHO, Tylenol Official, The Guardian, British Heart Foundation, and UK Mental Health) and with high numbers of views (greater than 25,000 views) on YouTube.com or Vimeo.com.

- The final list consisted of nine videos; three on each topic. Videos were checked by three researchers for suitability for the study. See Appendix A for the list of Data Videos used.

## 3.3 Data Collection Instruments

### 3.3.1 Demographics

The first part of the study asked demographic questions (e.g., age, sex, first language) followed by questions about participants' interest levels on the three health topics (i.e., physical activity, diet, and sleep).

### 3.3.2 Personality Traits

The next section assessed participants' personality traits. A version of the Big-Five Inventory with 10 questions [66] was used because speed was crucial due to the online nature of the survey, see Table 1. This version of the scale is widely used in personalized technologies [9, 63] that tailor their contents based on the users' personality and in which personality assessment needs to be quick. Although it is relatively short compared to the standard multi-item instruments, the 10-item version has been repeatedly examined and verified. According to Gosling et al. [33] it has "reached an adequate level" in terms of *predictive power* and convergence with full scales in self, observer, and peer responses. [1]

### 3.3.3 Perceptions of Own Health

Participants were asked to answer questions about their own diet, sleep, and physical activity in general (e.g., "Generally speaking,

---

[1] According to Google Scholar search, this 10-item scale has been cited in 2902 articles at the moment of writing; Sept, 2020

---

Table 2: Negative Affects Question Items

| Please read each statement carefully, and select the appropriate answer that best describes how you feel **right now.** |
| --- |
| I feel anxious.
I am relaxed. (R)
I am worried. |

*(R) = item is reverse-scored.*
*A Likert scale (1: Not at all to 8: Extremely) is used.*

I am physically active"), using a 7-point Likert scale (1; Strongly Disagree to 7; Strongly Agree).

### 3.3.4 Affective State Self-Reports

Participants' negative affects, focusing on their worries, anxiousness, and (not being) relaxed, were assessed using three questions, four times; first, prior to the exposure to any of the videos (for a baseline value) and right after viewing each video, to examine the affective influence of the video. We used an 8-point Likert scale to report the affect intensity (1 = not at all; 8 = extremely; see Table 2). We were inspired by [80] and followed their approach of not having a mid point in the scale as we focus on negative affects.

We chose to focus on negative affects for two main reasons. First of all, as noted earlier, the model of behaviour change [34] suggests that feeling worried or at risk is the first step towards attitude change. Second, the majority of health Data Videos that we looked at contained unpleasant facts and threatening messages.

### 3.3.5 Persuasive Potential Questionnaire (PPQ)

This study explored the effect of Data Videos at the perceptual level as a preliminary step in investigating Data Videos. More specifically, we focused on participants' motivation and willingness to change their behavior as opposed to their actual behavior change. While exploring behavior changes would have been useful, it requires a longitudinal study which was not possible with our current restrictions due to the pandemic. Therefore, to measure the potential of Data Videos, the Persuasive Potential Questionnaire (PPQ) [54] was adopted and adjusted to fit our context. PPQ is a subjective measurement tool that allows us to assess the potential of a persuasive system. The scale is composed of 15 question items, reported using a 7-point Likert scale (1; Strongly Disagree to 7; Strongly Agree); grouped under 3 dimensions: 1) individuals' susceptibility to persuasion (SP), 2) the general persuasive potential of the system (GPP), which measures the participants' perception of the system's ability to persuade, and 3) the individual persuasive potential of the user (IPP) which measures participants' assessment of the persuasive potential of a system they tried; (See Table 3). We did not include the IPP dimension as this set of questions (e.g., "I think I will use such a program in the future") are irrelevant to our research goal. Thus, we used the first two dimensions of PPQ. Since the SP dimension measures personal traits that are independent of the system, we asked participants to respond to it prior to the video viewing. Participants responded to the GPP questions after the video viewing to report their perception of the potential persuasive ability of each video.

## 3.4 Overall Study Progression

First, participants answered demographic questions and SP questions in Table 3, followed by the 10-item personality measure (See Table 1).

Participants then watched three Data Videos and answered questions after each. The videos covered the three health topics, randomly selected from the sets of three videos per topic (see Figure 2). The

Table 3: Adjusted Persuasive Potential Questionnaire

| | | |
|---|---|---|
| **SP** | 1 | When I hear others talking about something, I often re-evaluate my attitude toward it. |
| | 2 | I do not like to be influenced by others. |
| | 3 | Persuading me is hard even for my close friends. |
| | 4 | When I am determined, no one can tell me what to do. |
| **GPP** | | **I feel that...** |
| | 5 | the video would make its viewer change their behaviors. |
| | 6 | the video has the potential to influence its viewer. |
| | 7 | the video gives the viewer a new behavioral guideline. |

A Likert scale (1: Strongly Disagree to 7: Strongly Agree) is used

| 3 Physical Activity videos | 3 Sleep videos | 3 Diet videos |
|---|---|---|
| PA Video 1 | Sleep Video 1 | Diet Video 1 |
| PA Video 2 | Sleep Video 2 | Diet Video 2 |
| PA Video 3 | Sleep Video 3 | Diet Video 3 |

Figure 2: We had 9 videos in total: 3 videos per topic. Each participant watched 3 videos in total, 1 video on each topic. (e.g., Diet Video 2, PA video 1, then Sleep Video 2). Thus, the order of the topic and the selection of the video within each category were randomized.

order in which the topics were presented was also randomized for two reasons: 1) to avoid any priming effect that might occur due to the topic relevance to the participant, 2) to cancel out potential effects associated with features of each video. Participants were given full control to replay or pause the video. Our instruction made it clear that the participants could not skip to the next section (i.e., question) unless sufficient time (i.e., the length of the video) had elapsed.

After watching each video:

1. Participants answered the three Affect-related questions. This helped us to capture participants' affective state influenced by the video (see Table 2).

2. Participants answered three questions regarding their appraisal of the video content (Novelty, Clarity, and Usefulness of the information; e.g., "The information provided by the video was useful to me") using 7-point Likert scale (1: Strongly Disagree to 7: Strongly Agree). Their overall liking of the video was also assessed.

3. Participants completed the questions for the General Persuasive Potential (GPP) of the video (see Table 3).

4. Finally, participants indicated if they had any health issues that would prevent them from following the video's advice.

After completing these four steps for each video, participants were asked to solve a one-minute, 12-piece jigsaw puzzle. This step was created to help participants neutralize their affective sate between videos by focusing on a task. After the puzzle, participants repeated the four steps for the next video. In total, each participant watched three videos and did two puzzles (one puzzle between 1st and 2nd video, and another puzzle between 2nd and 3rd video). After the final video, participants were directed to a final page that thanked them for their participation, and their work was submitted for review and payment following M-Turk standard practices.

### 3.5 Hypotheses

In this study we had the following five hypotheses:

$H_1$: Watching Data Videos will increase participants' negative affects.

$H_2$: There are correlations between Personality traits and Negative Affects.

- Neurotic people tend to be anxious and more likely to feel threatened by ordinary situations [78], they could experience more negative affects.

- Extroverts and people open to experience are characterized by their happiness and optimism. Thus, we do not expect correlation between negative affects and these traits (i.e., lower susceptibility to threatening messages).

- Conscientious individuals tend to be cautious. Thus, they could become worried about their own health after watching the video. Alternatively, they might be inclined to process threatening information cognitively, and as a result, they might not experience intense negative affects.

$H_3$: Negative affects predict potential attitude change, measured by PPQ.

$H_4$: There is a link between personality traits and potential attitude change.

$H_5$: There is a link between video appraisal factors and potential attitude change.

## 4 RESULTS

On average, participants took 26 minutes to complete the study. Data-fitting assumptions for each analysis were checked and non-parametric options were used whenever appropriate.

### 4.1 Participants

We recruited participants ($N$ = 102; 68 Males, 33 Females, and one participant preferred not to say) with ages ranging between 21 and 70 ($M$ = 37.29, $SD$ = 12.01). 60% of the participants identified themselves as white, 20% preferred not to mention their ethnicity and the rest were Hispanic, Black, Asian, and American. 100 participants reported their first language was English, 83.3% of them had an education level higher than Bachelor's Degree.

### 4.2 Data Quality Control

A verifiable (i.e., Gotcha) question was included in the survey. This question was designed to be readily solvable as long as the participants read the question ("How many words do you see in this sentence?"): 78 valid cases remained for the analyses. When appropriate, we further filtered out responses when participants responded "Yes" to the following question ("I have health issues that prevent me from following the advice provided in the video") in each of the three topics (i.e., Physical Activity, Sleep, Diet). [2]

### 4.3 Data Videos and Negative Affects

To explore $H_1$, a Wilcoxon Signed Ranks Test explored whether viewing of Data Videos influenced the levels of participants' negative affects; prior to the video viewing ($Mdn$ = 2.67) and after the first video viewing ($Mdn$ = 2.33). Contrary to our expectation, participants' negative affect was not heightened even after they viewed a video ($Z$ = -.101, $p$ = .919). Note the video order was randomized and thus, this lack of effect cannot be interpreted as a result of one specific video.

---

[2] This choice was made to reduce potential confounds (i.e., the participants might not be willing to change their attitude in response to the video because of their health issues). Four participants responded "Yes" after watching a video related to physical activity, and four different participants responded "Yes" after watching a video related to sleep, and finally, three participants responded "Yes" after watching a video related to diet. A pairwise deletion method was applied to this selection throughout the analyses.

## 4.4 Personality Traits and Negative Affects

To examine $H_2$, correlations between each personality trait and negative affects were explored. Negative affects were positively correlated with neuroticism, $rho$ (78) = .594, $p <$ .001, and negatively correlated with conscientiousness $rho$ (78) = -.363, $p$ = .001; no other traits were correlated with negative affects.

## 4.5 Negative Affects and GPP

We examined the link between negative affects and potential attitude change ($H_3$). For the analysis, we computed an index for Affective Responses. First, Chronbach's Alphas were checked ( $.73 \leq \alpha \leq .82$) for participants' affective responses (anxious, relaxed, and worried) per topic (Physical Activity, Sleep, and Diet). [3] Since the alpha levels satisfied our standard (.70) [60], the mean of these three items was computed. Then the correlations between these means for each topic were also investigated. They were all significantly correlated ($.810 <rhos <.830, ps <.001$; see [4]). This implies that if a participant's affective response was negative from viewing one video, it was likely that they experienced negative affects from viewing other videos as well (i.e., implied underlying personal tendency). Thus, the mean across all the topics was used as an index for Negative Affect. The index for GPP was also created in the same manner. Chronbach's alphas ranged between .81 and .91 per topic. We further checked whether GPP for one topic (e.g., Physical activity) was correlated with the GPP for other topics (e.g., Sleep and Diet). They were significantly correlated with each other ($.555 <rhos <.719, ps <.001$)[4], and the mean of scores across all the topics was used as a GPP index. We explored whether overall GPP could be predicted by negative affects with a linear regression analysis. Negative affects predicted GPP, $F$ (1, 76) = 4.056, $p$ =.048, $R^2$ change = .051, $\beta$ = -.225.

## 4.6 Personality Traits and Potential Attitude Change

To explore $H_4$, first, we explored the link between personality traits and individuals' susceptibility to persuasion (SP). Since Chronbach's Alpha for the four SP items was .60, we removed the first item (See Table 3) based on its low correlation with other items ($ps \geq$ .248). Thus, the mean of these three items (2, 3, and 4, see Table 3 ) was used to create an index of SP (Chronbach's Alpha = .79; [60]).[5] Agreeableness was positively correlated with SP, $rho$ (78)= .26, $p$ = .047. No other links were found. The more agreeable participants were, the more susceptible to persuasion they were and vice versa.

Next, linear regression analysis explored traits as predictors of GPP index using stepwise method. Neuroticism was the only predictor of GPP, $F$ (1, 76) = 8.179, $p$ = .005, $R^2$ change = .097, $\beta$ = -.306. When individuals are highly neurotic, it was harder to achieve higher GPP. Based on this, we turned to focus on neuroticism to explore what it does to viewers' cognitive processing. Thus far, we have found that neuroticism is correlated with negative affects. Now, we further explored to see whether neuroticism predicted participants' general cognitive tendency even before they had watched the videos. For this, neuroticism was used as a predictor while participants' own health perception for each topic was entered as a dependent variable. Participants' health-related perception for all the topics were predicted by neuroticism at significant level; Physical Activity, $F$ (1, 65) = 5.04, $p$ =.028, $R^2$ change = .072, $\beta$ = -.268; Sleep $F$ (1, 64) = 6.37, $p$ = .014, $R^2$ change = .091, $\beta$ = -.301; Diet, $F$ (1, 65) =

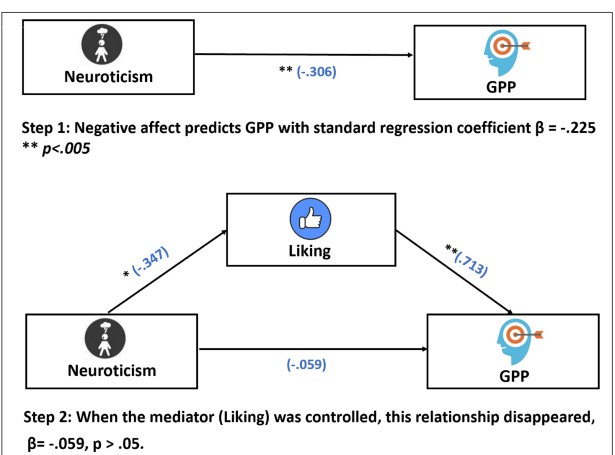

**Step 1: Negative affect predicts GPP with standard regression coefficient β = -.225**
** p<.005

**Step 2: When the mediator (Liking) was controlled, this relationship disappeared, β= -.059, p > .05.**

Figure 3: Liking as mediator between neuroticsim and GPP. Neuroticism originally predicted GPP (p <.005; Step 1). However, when we controlled for the mediator (Liking), this relationship disappeared (p > .05; Step 2). This indicates one potential way designers could incorporate personality traits in persuasive design of Data Videos. *$p$<.05. *$p$<.005.

24.87, $p <$.001, $R^2$ change = .277, $\beta$ = -.526. We suggest this could be explained by the link between thinking style and neuroticism discovered by Zhang [81] where researchers found that neuroticism was linked to a conservative and risk-averse thinking style. In line with their findings, participants who scored high on neuroticism also revealed rather conservative views about their own health status and judged the extent of a videos' persuasiveness rather conservatively.

Finally, we explored the potential explanation of underlying dynamics of how neuroticism predicted GPP in our data. Inspired by previous findings, we hypothesized that liking of the video could be a mediator of the link between neuroticism and GPP: Neurotic people might not like the video (i.e., judging the video conservatively), and that could, at least partially, explain why their potential behavior change is not expected. To explore this, we followed Baron's mediation analysis [12] again, and mediation effect was found. Although neuroticism originally predicted GPP, $\beta$ = -.306, t(77) = -2.86, $p$ = <.005 (See Step 1 in Figure 3), this effect disappeared when liking of the video was controlled, $\beta$ = -.059, $t$(77) = -.781, $p$ =.437. This result helps designers see how they should consider personality differences in their Data Video development (i.e., personalization). We found that it is particularly challenging to persuade individuals who are highly neurotic. To guide them to alter their willingness to change their health-related behaviors, then, providing Data Videos that neurotic individuals *like* was a key.

## 4.7 Content Appraisal

For exploratory purposes, we explored how participants perceived the content of the videos (Information Novelty, Information Clarity, Information Usefulness, See Figure 4). Their content evaluation of one video correlated with the evaluation of the rest ($.328<rhos<.700, ps <.005$). Finally, we conducted a regression analyses to explore how content appraisal variables and personality traits variables could predict GPP together. We entered GPP as a dependent variable while Big Five traits and three content appraisal items were entered as predictors using the stepwise method. Only two predictors remained in the model, which explained approximately 67% of the variability in GPP (1; Information Usefulness, $\beta$ = .718, $p <$.001 , 2; Neuroticism, $\beta$ = -.143, $p$ =.036), $F$ (2, 75) = 74.87, $p <$.001, $R^2$ = .666 (See Figure 4).

---

[3] One item ("I am relaxed") was reverse coded for the index computation.

[4] This implied another underlying personal tendency; if a participant perceived high levels of general persuasiveness in one video, they perceived higher levels of general persuasiveness in other videos as well

[5] Correlations between personality traits and SP were explored instead of regression. This was because, conceptually, personality traits should be related to SP but they both represent individual's attributes. Hence, we did not suspect one can predict the other.

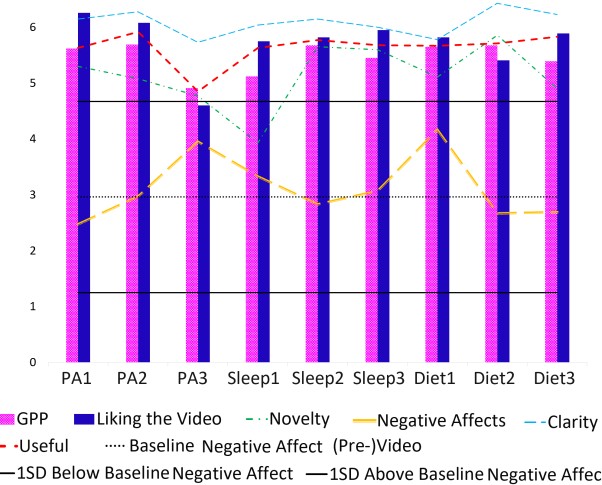

Figure 4: GPP, Negative Affect, Topic Interest and Content Appraisal means per video. Only Usefulness predicted GPP.

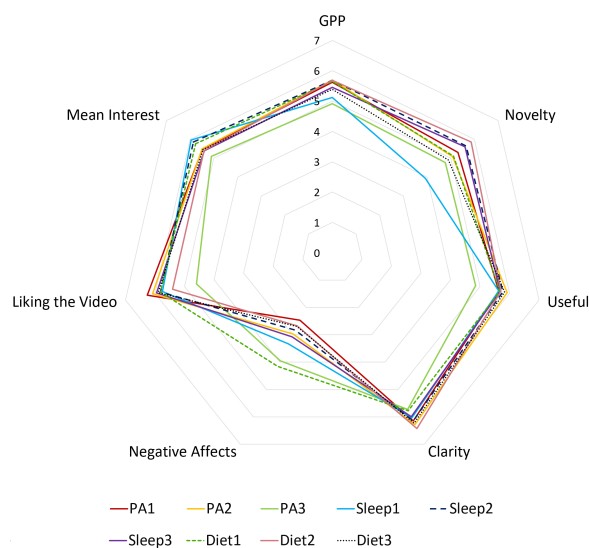

Figure 5: General trends by individual video. Overall, selected videos induced generally comparable effects, implying that although we did not create videos for the study, we sufficiently controlled for the basic quality of the videos.

## 5 DISCUSSION

Due to the nature of common health Data Videos which regularly contain fear inducing messages, in this study, we explored negative affects (i.e., anxiety, worries, and not being relaxed in response to those messages) specifically about physical activity, sleep, and diet. While we did not find evidence to show Data Videos increased the levels of negative affects, the levels of negative affects predicted general persuasive potential. These findings suggested that levels of negative affects were not influenced by our stimuli at least at a significant level. Further, and importantly, neuroticism predicted general potential persuasiveness. Specifically, when individuals scored high in neuroticism (in comparison to those who scored low), their willingness to reconsider their health-related behaviors was rather low. How do we tackle this challenge then? Can we influence individuals' perception when they they score high on neuroticism? Is there a way to persuade neurotic individuals? Our mediation analysis shed some lights on this neuroticism-persuasiveness link. Our results showed that the neuroticism-persuasiveness link disappears when we remove the effect of individuals' liking of the videos. That is, when we controlled for their liking of the videos, neuroticism did not predict general persuasiveness anymore. This indicates that, in persuading highly neurotic individuals, focusing on improving likability of the Data Videos should be beneficial. Then, understanding viewers' preferences/taste based on their personality type could be fruitful for Data Video designers in achieving higher persuasiveness, at least at a perceptual level.

Additionally, we were able to explore the contents of the videos to find general potential guidelines in designing health-related Data Videos. While our exploration is limited to three aspects of the content (Usefulness of the information, Importance of the information, and Clarity of the information), we were able to find the significance of perceived usefulness of the information in predicting general attitude change. Moreover, our model with neuroticism and perceived usefulness of the information together explained the 67% of the variability in general potential attitude change at a significant level. This indicates the importance of consideration of both sides, the audience (e.g., personality traits) in conjunction with the video content itself, to promote attitude change effectively. Improving "Perceived Usefulness" of health-related Data Videos appears to be one of the general rules in developing health-related Data Videos.

Altogether, we were able to find potential means to approach individuals who are high in neuroticism with Data Videos. While they might be harder to persuade, delivering Data Videos that they *like* might allow us to get us closer to the goal of health-related Data Videos. At the same time, this finding enforces the argument of personalization of technology. Our results indicated personalizing Data Videos at least for those who score high on neuroticism could be useful: Personalization of health-related Data Videos could be, in fact, essential to alter their viewers' perception effectively. On the other hand, regardless of the personality of the viewers, focusing on the improvement of perceived usefulness in Data Videos could enhance their potential of general persuasiveness. In sum, we were able to find means to improve the potential of general persuasiveness by targeting both general population and specific individuals.

## 6 LIMITATIONS AND FUTURE WORK

Due to the COVID-19 pandemic, we were not able to conduct the study in the laboratory setting: instead, we used M-Turk. Although M-Turk gave us the opportunity to recruit participants in a short period of time at a relatively low cost, it compromised the controllability of our study. For the same reason, we only relied on participants' self-reporting of their affective responses, and were not able to use any physiological measurements to verify their reported affects. Future studies with physiological measurements will be useful to validate our findings. Furthermore, the study was limited to negative affective responses related to anxiety. We acknowledge that the current time of the pandemic might have affected our results, as participants may be experiencing higher levels of general anxiety than usual. Moreover, the investigation of positive affects would also be useful. Examining positive affects such as excitement or hope, along with negative affects, will improve our model further. Finally, stimuli used in this study were not created for the purpose of this study. Instead, we selected nine existing videos systematically. While we might have relatively lower control in our stimuli (e.g., voice over by male vs. female, use of animation, font types), we chose to use existing data videos to help maximize the generalizability of our results. The general data trends by individual video (Fig 5) shows that central quality of the videos were consistent across videos, confirming that our video selection was successful.

# 7 CONCLUSION

We found some evidence that neuroticism is an important trait to be considered towards personalization of persuasive technology, at least for health-related Data Videos (Physical Activities, Sleep, and Diet), when motivating the viewers is the goal of such videos. When individuals score high in neuroticism, they are harder to persuade, but there is a potential solution via personalization. Our results further indicated that negative affects (feeling of anxiety/worry) would not aid in the potential to attitude change. It is worth mentioning our results implied that those negative affects could be attributed to the participants' traits in the end. Altogether, these findings encourage us to consider personality traits in designing Data Videos. As for video attributes, we found that perceived information usefulness was a key factor influencing the persuasive potential of the video. While our model with neuroticism and perceived usefulness of the information predicted general potential for attitude change, there are numerous other content related factors to be explored in future studies (e.g., video length). With our results, we would like to conclude that consideration of personality traits along with attributes related to the video content would be beneficial in developing Data Videos for promoting health-related attitude change.

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
