# OpenReview forum: "Can Data Videos Influence Viewers’ Willingness To Reconsider Their Health-Related Behavior? An Exploration With Existing Data Videos"
_graphicsinterface.org/Graphics_Interface/2021/Conference — Submitted to GI 2021_

### Official Review · AnonReviewer1 · 2021-01-12
**Is fear an effective motivator for better health?**

**Rating:** 3
**Confidence:** 5

**Review:**

This is a well-written paper that explores an important problem, namely how can we use technology to foster better health behaviors. The authors conduct a mechanical turk study on ~100 participants (76 useful participants after filtering questions) to determine whether short health videos foster an increased level of fear or concern regarding health related issues and, alongside this, whether participants' behaviors will change.

At the outset, I question a fundamental assumption of this paper, best expressed by the authors in paragraph 2 of the paper:
"As behaviors are normally adjustable, people suffering from
behavior-induced health issues would benefit if they understood how
their seemingly minor behaviors are tied to their serious medical
issues. Further, they could be motivated to reevaluate their behaviors
if they were guided with concrete behavioral suggestions on how
to improve their health."

Let me unpack this a bit. Is it truly the authors' contention that people do not understand the consequences of negative health behaviors? I, personally, don't think this is the case. I should exercise more, eat fewer calories, preference whole grains, fruits and vegetables, avoid sugar and saturated fat, drink less coffee ... The list goes on. And I *know* the negative consequences of these actions. But I persist -- not because I don't want to change but because the immediate temptation of a dessert outweighs the long-term thought process of fear of health-related issues. Or the immediate need of a coffee outweighs the fact that I've had five so far today.

Essentially, to my mind, the underlying assumption that motivated exploring these health issues is faulty, and this, in turn, results in a set of results from which it is hard to draw anything useful. As the authors note, negative affect wasn't impacted by showing videos. Why? Because everyone already knows the negative consequences of much of the behaviors explored. Stated another way, if I think about this from, for example, Fogg's Behavior Model for persuasive technology, what the authors are trying to do is increase motivation, and to do this through negative emotional responses (specifically fear). However, the challenge is that the domain chosen -- health related interventions -- is one where the viewer already knows the consequences, undoubtedly fears them (at least in the abstract) and misses, instead, the ability to easily change behavior and/or some trigger to encourage them to make a better decision in the moment (again borrowing from Fogg's model).

One thing that I try to do in reviews is to highlight where and how I think the authors could re-interpret their data to enhance their results. Unfortunately, this is one case where I simply don't think it's possible to pull anything from the data set. The assumption of the authors was that people don't know the long term consequences of their behavior, they don't fear them enough, but here I think they do. The authors results even support this assumption -- show them the video and their fear is not increased; it actually is reduced. I think the problem is that -- even with the pre-existing fear -- there is no way to adjust motivation through that fear. Instead, we need to look elsewhere, to questions such as:
- Can we motivate people by making positive behaviors fun?
- Can we lower the barrier to act healthier (e.g. make it easier to exercise, add bike lockers at work, simplify showering and changing, subsidize public transit, offer free gym memberships, provide healthy lunch options at work ...)?
- Can we remind people in a timely fashion to pursue those positive behaviors (e.g. daily exercise reminders, break reminders, etc.)?

So ... I commend the authors for the domain they are working in. However, I feel that the intervention chosen is simply the wrong one. On the positive, this was a relatively low cost study to run -- select some videos, show them to MTurkers, find the negative results -- so the cost of the false start was low.

Finally, I spent some time thinking about whether or not, even with negative results, it might be worthwhile to publish this paper as a negative results paper, but, in this case, I don't see that path forward. Negative results are interesting when they are surprising and can better inform future work. Here, we've seen advertisements ad infinitum on the negative health aspects of x, y, or z, yet we persist. Because that the negative is insufficient is known.

---

### Official Review · AnonReviewer3 · 2021-01-14
**Data videos need to be personalized**

**Rating:** 4
**Confidence:** 3

**Review:**

The paper studies publicly available data videos addressing health-related behaviour. Through a M-Turk study, it ask if the videos increase negative affects and if the negative affects are linked with personality traits, can predict changes to viewers’ willingness to improve their behaviour. It further asks if the willingness can be predicted with personality traits and/or video attributes. The study finds that the perceived usefulness of the information presented in the videos along with neuroticism predicted our participants’ willingness. The authors also find that the neuroticism-persuasiveness link relates to the likability of the video.

More information is required on how the final 9 videos were selected. Authors mention “length, source credibility, information accuracy, etc” - can they explain this in more detail. This is especially important as finding relevant videos is an important challenge in a study such as this. I am also not completely convinced that the PPQ Persuasive Potential Questionnaire can be used to assess the potential of data videos. As the paper also mentions, the original PPQ was specifically created for assessing the persuasiveness of systems - it is derived from the TAM/UTAUT models. The study needs to motivate how it can use the PPQ to analyze persuasiveness of videos better. There is literature in communication studies on “Narrative Persuasion” that might fit this better, and I would encourage the authors to look at other sources that look at the persuasiveness in videos.

Besides these concerns, the study itself seems well-designed and the results seem robust. However, I have concerns that the final results are not novel enough. Yes, videos should be personalized, and yes, people high on neuroticism might be harder to persuade through negative videos. But the deeper question. which the authors don't seem to be considering, is whether negative videos are really persuasive in the first place. A separate study might use both negative videos and positive videos.

In summary , I feel like the authors need more details about the videos in the paper. How were they chosen? More details about what the content consisted of? How can we measure the persuasion of videos given the emotional and information content (and not use a related but not completely relevant measure, i.e. PPQ)?

On a side-note, the authors must reconsider using the Big-Five-10 questionnaire. It is fine as a base level test, but if the paper is delving deeper into how personality traits affect consumption, they need to look more questions/dimensions to truly isolate relationships.

---

### Official Review · AnonReviewer2 · 2021-01-14
**Paper is well written and study seems valid**

**Rating:** 6
**Confidence:** 3

**Review:**

This paper describes a study looking at the relationship between affect and behaviour change in data videos.

The paper is well written and there has clearly been much thought put into the study design and analysis. I am not an expert in the area or the methods used, however the study design and analysis seem to be valid. The conclusions drawn from the results seem a little tenuous, but somewhat convincing.

The main issue I see in this work is that it doesn't address the potential issues of using affect to sway opinion instead of fact. If a video is heavily relying on affect, I don't see how this can be classified as a 'data video'. The related work does mention differences between the two (central route vs peripheral route), but a more thorough discussion is warranted. I note that the authors removed videos with false claims from the study, so perhaps this issue can be safely abstracted from the current study. Nonetheless, it seems important to discuss the potential application for misuse of such an approach.

---

### Meta-Review · Area_Chair1 · 2021-01-15

**Recommendation:** Reject
**Confidence:** 5

**Metareview:**

This paper has reviews that range from a clear reject to possibly above the bar for acceptance.

Looking in detail at the reviews, the most positive reviewer, R2, while rating themselves as "fairly confident", acknowledges in their review that they are not expert in either the area nor the methods used. R3, in contrast finds information missing, and R1, the most negative reviewer, questions whether the intervention was the correct one. Overall, given the details provided in the review, I feel that this paper is, at this point, not yet ready for publication. Adding details suggested by R3 may improve the overall score of this work to the point where R1's review could be discounted. However, possibly not. R1 notes some significant shortcomings of this work, and, at the very least, an extensive discussion addressing these points should be provided.

However, taking reviewer scores together, my meta-evaluation indicates a reject on this paper.

---

### Decision · Program_Chairs · 2021-01-16

Reject